# NiAl-Cr-Mo Medium Entropy Alloys: Microstructural Verification, Solidification Considerations, and Sliding Wear Response

**DOI:** 10.3390/ma13163445

**Published:** 2020-08-05

**Authors:** Christina Mathiou, Konstantinos Giorspyros, Emmanuel Georgatis, Anthoula Poulia, Alexander E. Karantzalis

**Affiliations:** Department of Materials Science and Engineering, University of Ioannina, 45100 Ioánnina, Greece; ChristinaMathiou@hotmail.com (C.M.); kgiors64@gmail.com (K.G.); apoulia@cc.uoi.gr (A.P.)

**Keywords:** high-entropy alloys, solidification, sliding wear

## Abstract

A series of NiAl-Cr-Mo systems were produced and assessed as far as their microstructure and their sliding wear resistance is concerned. The NiAl content was kept constant and seven compositions of Cr-Mo were tested, namely, 40Cr-0Mo, 30Cr-10Mo, 25Cr-15Mo, 20Cr-20Mo, 15Cr-25Mo, 10Cr-30Mo, and 0Cr-40Mo. It was observed that most of the systems contained primary phases, eutectic microconstituents, and, occasionally, intermetallic phases as the outcome of peritectic reactions. The extent and the nature of all these microstructural features was proved to be affected by the Cr/Mo relative ratio, and an attempt was conducted in order to explain the microstructural features based on solidification and other related phenomena. It was observed that the increase of the relative Mo/Cr ratio led to a significant restriction/elimination of the eutectic microconstituent. The sliding wear response of the produced system seems to diverge from the classical sliding wear laws of Archard and is based on multiple factors such as the nature of the oxide phases being formed upon sliding, the nature and the extend of the intermetallic phases being formed upon solidification, and the integrity and rigidity of the primary phases—last to solidify areas interfacial region and the factors that may influence this integrity.

## 1. Introduction

During the last decades, NiAl-based alloys have been considered as potential candidates for high-temperature applications due to their enhanced properties at such extreme servicing conditions and possible substitutes for Ni-based superalloys [1]. However, the brittle nature of the NiAl intermetallic phase and their limited ductility, early enough, raised question on their possible modification so that their toughness and ductility can be improved. Towards this direction, research efforts were focused on the addition of refractory metals (Cr, Mo, Ta, etc.) within the NiAl matrix in order to overcome these drawbacks and their properties were, in most of the cases, evaluated in depth [2,3,4,5,6].

Within this frame, the addition of Cr within a NiAl intermetallic matrix has gained special scientific and research interest [7,8]. The potential substitution of Ni-based super-alloys, nevertheless, was not the only driving force for the development of NiAl-based system. The immergence of the new class of metallic materials, high-entropy alloys (HEAs), revealed another direction of the importance of the Ni-Al-Cr elemental combinations. Indeed, a wide range of this novel material class has been based, developed, and assessed on the Ni-Al-Cr core [9,10,11,12,13,14]. This intensive research effort clearly showed the necessity and the importance of the in-depth understanding of the microstructural characteristics and mechanisms related to them and phenomena of the NiAl-Cr system. Fortunately, for the research community, Tang et al. [15] presented an exceptional research work where, based on thermodynamic and kinetic calculations, they showed a thorough approach on the possible microstructural configurations of the NiAl-Cr systems for various Cr compositions. Similar thermodynamic calculations of the Ni-Al-Cr system, with very important and useful results, were also performed by Duprin et al. [16].

In parallel with the works focused on the addition of Cr in NiAl matrix, Mo has also attracted significant research attention, and various works have been conducted concerning the development and microstructural property evaluation of NiAl-Mo systems with very promising results [4,5,17,18].

By the development of HEAs point of view, a combination of Cr and Mo would be an interesting approach of gradually building medium to high-entropy systems. This combination of four elements (Ni-Al-Cr-Mo) has been examined in various research efforts. Chen et al. [19], for instance, studied the deformation and fracture behavior of directionally solidified NiAl-28Cr-6Mo eutectic alloy. They observed a eutectic lamellar microstructure with both the consisting phases being grown parallel and towards the same direction. The semicoherent nature of the eutectic constituent interface was responsible for the fracture behavior of this alloy. Whittenberger et al. [20] also examined a directionally solidified NiAl-31Cr-3Mo eutectic alloy and commented on the transition from lamellar to cellular configurations as a function of the growth rate. High-temperature compression tests did not reveal, in their work, any significant difference between the different eutectic patterns as far as the fracture toughness is concerned. Raj and Locci [21] proved that the different withdrawal rate significantly affects the microstructure of a eutectic NiAl-31Cr-3Mo alloy. Similar observations on the effect of solidification rates on the microstructure of Ni-33Al-33Cr-3Mo and Ni-33Al-31Cr-3Mo eutectic alloys were presented by Raj et al. [22]. Shang et al. [23] examined the microstructure and the room temperature fracture toughness of a directionally solidified NiAl-Cr(Mo) alloys. They observed that the withdrawal rate, the temperature gradient, and the Cr composition are the main factors that affect the microstructural characteristics and the fracture toughness of the produced alloys. These indicative research efforts, provide a first hint on the importance of the NiAl-Cr-Mo alloy systems. Peng et al. [24] with their thermodynamic calculations through CALPHAD provided a thorough insight on the possible microstructural characteristics and morphology in the NiAl-Cr-Mo systems. Their theoretical predictions were also validated by their experimental results.

Despite the fact that the microstructure and the mechanical properties of NiAl-Cr-Mo systems were examined through various research efforts, limited work has been conducted so far on the sliding wear response of these systems. The most systematic approach on this issue has been conducted by Guo and his colleagues [25] who examined the sliding wear response of various NiAl-Cr-Mo-based systems with and/or without the presence of other externally introduced reinforcing phases. However, this specific field is open for further experimental and research investigations.

The present work is part of a wider research effort to design and develop new high-entropy alloys around the fundamental core of the NiAl-Cr system. The authors have, at a first instance, investigated the basic NiAl-Cr core system, as far as the microstructural features and the related mechanisms are related, along with the alterations that small additions of Mo may cause [26]. In the present work, higher Mo additions (up to 40 at.%, no Cr) were selected maintaining, however, a constant Cr-Mo addition of 40 at.%. This compositional range was selected in order to examine the influence of the Cr/Mo ratio on both the microstructural features and the sliding wear response of the produced alloys. The objective of this approach is to establish compositional ranges of Ni-Al-Cr-Mo for optimum medium entropy alloy. The primary target was to establish microstructural morphologies where the eutectic microconstituent is significantly eliminated and if possible vanished. Vanishing of the eutectic phase will lead to simple microstructures that may permit more flexible and precise control of the system properties. This will permit the transition to the next level of next elemental addition (primary work on W additions has already been undertaken), in order to develop new high-entropy alloys of simple microstructures.

## 2. Experimental Procedure

High-purity elemental powders were mixed in the intended atomic ratio in order to produce various Ni-Al-Cr-Mo alloys. Approximately, 5 g of raw materials, with purities higher than 99.5%, were subjected to uniaxial compression in order to make green pellets with targeted (nominal) composition according to Table 1. The produced green pellets were arc melted at least 5 times in order to ensure that all raw materials were well mixed in liquid prior to solidification. The samples were flipped for each melting cycle in order to improve chemical homogeneity. The arc current was kept constant at 120 A in all melting and remelting steps. The melting was performed on a water cooled copper-based mold. Cooling conditions associated with the final microstructure are also mentioned in other experimental efforts [27,28]. The actual composition was checked using EDS mapping in the SEM in at least 3 different locations of lower magnification and all fell within the measured compositions presented in Table 1.

The microstructure of the alloys was studied in their as-cast condition. The metallographic specimens were mounted in Bakelite, abraded on SiC papers up to 2400 grit, and then polished with a 3 μm diamond suspension. The polished specimens were etched using aqua regia. Alloy’s microstructure was analyzed with the use of a scanning electron microscope (SEM) (JEOL 6510 LV, Tokyo, Japan) equipped with both backscatter electron (BSE) and energy-dispersive spectroscopy (EDS) detector of X-Act type by Oxford Instruments (Oxford, UK). X-ray diffraction (XRD) patterns were obtained using a D8 ADVANCE diffractometer (Bruker AXS GmbH, MA, USA), operating with CuK_α_ radiation (λ = 1.5406 Å) and a secondary beam graphite monochromator. Samples were scanned over an angular 2θ range from 20° to 120°, in steps of 0.02° (2θ) at a rate of 2 s per step (overall run duration 10,000 s).

Vickers microhardness (HV_1_) was measured using a 136° diamond pyramid for 5 s of load (1 kg) application. Sliding wear tests were performed using a ball-on-disk apparatus (CSM Instruments, Zurich, Switzerland) at ambient temperatures with relative humidity around 70%. The externally applied load was 5 N. The overall sliding distance was 1000 m, and the sliding speed was 10 cm/s. A 100Cr6, 6 mm steel ball was used as a counter body material. All specimens were acetone cleaned and their weight was measured. Tests were interrupted every 200 m, and the specimens were reweighted. Wear tracks were examined by SEM-EDS analysis. Three independent runs were carried out for each individual alloy system.

## 3. Results and Discussion

### 3.1. Parametric Model Initial Predictions

A detailed analysis of the parametric models for the prediction of the microstructural features of HEAs is given in Appendix A.

Table 2 presents the various calculated parameters for the different parametric models for both the nominal and the actual compositions of the produced systems. It is important at this stage to comment on the predictions of these models. According to the Zhang et al. [29] parametric model, the values of parameter δ are below 8.5, which is the proposed upper limit, and so are the ΔH_mix_ values. On the contrary, half of the systems show values of ΔS_mix_ outside the range proposed for single-phase solid solution (30Cr-10Mo, 10Cr-30Mo, 0Cr-40Mo, 40Cr-0Mo, respectively), which is a strong indication for potential phase segregation in their case. In the case of the geometric factor γ proposed by Wang et al. [30], all systems show values below 1.175, which is the upper acceptable limit. In the King et al. [31] model on the other hand, despite the fact that it provides values of the δ parameter within the accepted ranges, the parameter Φ is, in all cases, significantly lower than the value 1.0, which is the lowest acceptable limit, and as such, strong tendency for phases segregation for all systems is expected. For the Yang and Zhang [32] model, values of δ and Ω are within the accepted limits, with the exemption of alloy (0Cr-40Mo), where Ω is less than 1.1 proposed by the model as the lower limit. One of the most important models, which is especially related to the present effort, is the model of Troparevksy et al. [33] that deals especially with the thermodynamic tendency for intermetallic phase formation. As in the present work, the NiAl intermetallic phase is one of the major constituents of the produced alloys, the predictions of the Troparevksy el al. [33] model gains extra importance. As such, the model predicts the inevitable formation of NiAl intermetallic phase since it possesses an extremely low ΔH of formation value (−677 meV/atom), which is by far outside the proposed −232 meV/atom < ΔH_f_ < 37 meV/atom range for single-phase stability. Finally, the model of Senkov et al. [34], which also pays special attention on the formation of intermetallic phases, clearly depicts that the formation of intermetallic phase (NiAl in the present case) is strongly expected.

A possible concluding remark, as the parametric model predictions are concerned, is the fact that there is no consistent tendency for either a single-phase solid solution or multiple-phase formation. The actual microstructures do verify the presence of multiple phases, as it will be discussed in the following paragraph. This skepticism on the parametric model prediction validity, which has been expressed by many researchers (Pickering et al. [35], Karantzalis et al. [36,37,38]), should, however, by no means reduce their importance as they do comprise powerful tools in the process of designing new alloys and systems of controllable microstructures.

### 3.2. Microstructural Features

Figure 1a–g presents the microstructures of the different alloys produced in the present effort, and Table 1 provides the EDS point analysis data associated with the different phases being present in each individual alloy. More specifically, according to Figure 1a, alloy A consists of primary dendrites and a eutectic microconstituent. It is also evident that the eutectic constituent morphology varies from a central (hive-like) configuration to a more lamellar-like structure towards the outer regions, close to the primary dendrites. EDS analysis (Table 1) shows that the primary dendritic phases are rich in Cr, whereas, within the eutectic phase, the light areas are rich in Cr and the dark phases are rich in Ni and Al.

Alloy B, according to Figure 1b, consists of a primary dendritic phase and a eutectic microconstituent. A gray phase parametrically of the primary dendrites can be distinguished. Based on the data of Table 1, the primary dendritic phase is rich mainly in Cr followed by Mo, with some Ni and Al being dissolved, whereas the dark phase of the eutectic area is mainly enriched in Ni and Al. The perimetric dendritic halo is rich in Mo and Ni. The eutectic phases seem to follow an almost typical lamellar eutectic growing mode varying from more refined configurations at the core of the eutectic areas, towards coarser modifications at the periphery of the eutectic regions, close to the interface with the primary dendritic phase boundaries.

By increasing the Mo content (alloy C), the microstructure is lightly modified by means that the eutectic phase is reduced and the dark interdendritic phase is expanded. EDS analysis reveals that inside the core of the primary dendrites, the Mo content is increased and the Cr content is decreased. The dendritic halo is rich in Ni, as in the previous case, and the dark phase is rich in Ni and Al. Despite the reduction of the eutectic area, its morphology is still of typical lamellar configuration.

According to Figure 1d (alloy D), further increase of the Mo content leads to further distinguished microstructural alterations. The primary dendrites persist to exist, yet the halo areas are more expanded. The dark phase areas are also increased and the eutectic areas are severely constrained, and, wherever present, their morphology starts to fade from the typical lamellar configuration. EDS analysis shows that the Mo content is significantly increased within the primary dendrite cores and the dendrite halos, whereas the dark areas are still rich in Ni and Al, practically unchanged compared to the previous cases.

In the case of alloy E, where the Mo content reaches the values of 25 at.%, the microstructure is further modified. The extent of the peripheral halos is increased as the extent of the dark phase also does. It is worth noticing, however, that the eutectic morphology is almost eliminated. Table 1 data show that the content of the primary dendrite cores and the halo areas is further increased, whereas the dark areas are constantly rich in Ni and Al.

Further increase of the Mo content, alloy F in Figure 1f, does not seem to considerably alter the morphological features, compared to the previous case.

Alloy G (no Cr addition), Figure 1g, appears to consist of two phases: a light phase rich in Mo and a dark phase rich in Ni and Al. However, limited areas of eutectic microconstituent can also be observed.

Apart from the various EDS analysis, XRD measurements complement the characterization of the various phases being formed in the examined alloys (Figure 2). It can be observed that the NiAl-40Cr alloy consists basically of two BCC phases, those of NiAl (B2) and Cr (A2), as expected. The introduction of Mo leads to the formation of two intermetallic phases (AlMo_3_ and MoNi). AlMo_3_ seems to vanish for high Mo additions (over 20%Mo), whereas MoNi persists for even higher Mo concentrations. The NiAl-40 Mo alloy contains practically Mo and NiAl phases. Another important observation is the fact that from alloy A to alloy G, there is a gradual shift of one of the BCC phases, that of Cr, from Cr in the case of alloy A to Mo in the case of alloy G, a fact that is also expected. Despite the fact that TEM examination could provide more details on the peritectic halos formation, such examination, as adopted in other experimental efforts [27,28], was not adopted since the SEM analysis provides a significant hint of the peritectic reaction progress.

The first important observation to be noticed is the fact that the microstructural features of each individual alloy system produced in the present effort and presented in Figure 1, are in agreement with the calculated partial liquidus surface of the NiAl-Cr-Mo system proposed by Peng et al. [24] and presented graphically reconstructed in Figure 3, especially as far as the eutectic phase presence and extend in concerned. According to Figure 3, as the Mo content increases, the overall system composition continuously diverges away from the eutectic liquidus line and, as such, the tendency of eutectic constituent formation gradually diminishes. Indeed, as the alloy microstructure show, in alloys with 20, 25, and 30 at.% Mo, respectively, the significant reduction of the eutectic regions becomes profound, which, at last, are practically vanished in the 40 at.% Mo system. On the other hand, as the Cr content increases (alloys with 25, 30, and 40 at.% Cr), the system compositions converge towards the eutectic liquidus line, and as such, the eutectic regions should gradually increase and dominate, a fact that is clearly verified by the obtained microstructures of these systems in the present effort. It has to be mentioned, however, that this is a general speculation on the microstructural features and that more detailed approach on the various phases formed in each case will be presented in a following paragraph.

### 3.3. Solidification and Phase Formation Considerations

It was shown in Section 3.2 that in overall, the microstructures of the produced systems follow the trends and the predictions of the work of Peng et al. [24]. It is important, nevertheless, to examine in more details, the various phases formed in each individual alloy and to formulate a potential solidification and microstructural evolution sequence.

#### 3.3.1. Alloy A:NiAl-40Cr

Since this alloy contains no Mo, an approach to explain the various microstructural features should be passed on the partial binary eutectic system. As such, according to the NiAl-Cr pseudobinary phase diagram proposed by Tang et al. [15] and Duprin et al. [16], this composition corresponds to a hypereutectic alloy, which subsequently means that its microstructure should consist of the primary A2 (Cr) phase and a eutectic microconstituent. Interestingly, the microstructure seems to follow this general pattern, however, two phases seem to grow primarily and prior to the eutectic constituent: a dark phase and a light phase that either grows peripherically to the dark phase or as an interdendritic phase. A line scan EDS analysis (Figure 4) from the center of the dark phase and outwards reveals that the dark core is rich in Cr (an A2 phase), whereas the light phase is rich in Ni and Al (a B2 phase). In order to explain this sequence of events other factors such as recalescence and undercooling should be taken into consideration. According to the predictions of Tang et al. [15], two important temperatures can play vital role in the final phase formation: the T0-B2 and T0-A2 temperatures. T0-A2 and T0-B2, according to Tang et al. [15] and Laughlin and Hono [39], are defined as the maximum temperature at the interface, where partitionless (a limited case where the concentration of the solid is equal to the concentration of liquid at the interface) solidification can occur. For this specific composition (40 at.% Cr), Tang et al. [15] show that T0-B2 is considerably higher than the T0-A2 one. The resulted microstructure shows, hence, that the expressed—at the initial solidification stages—undercooling falls below the T0-A2 phase, and as such, the A2 phase starts to nucleate and grow in a partitionless mode. As soon as a critical mass of A2 is formed, recalescence and subsequent heat release raises the temperature above the T0-A2 phase and below the T0-B2 phase, establishing conditions for the development of growth of the B2 phase as a second stage. Practically, the expressed undercooling established conditions of the so-called decoupled growth leading to a sequential development and growth of A2 and B2 phases in a partitionless mode. Similar phenomena of decoupled growth have also been reported in other experimental efforts dealing with regular and anomalous eutectic solidification [40,41,42,43,44,45,46]. It is also important to notice, based on EDS data of Table 1, that the Cr dissolution within this primary B2 phase is at contents higher than those predicted by the pseudobinary eutectic NiAl-Cr phase diagram proposed by Tang et al. [15] and Duprin et al. [16]. These higher Cr contents are a strong indication of a more intensive solute trapping effect.

Once the primary B2 is formed, further heat release due to recalescence causes temperature increase at the surrounding liquid above T0-B2, establishing conditions for eutectic undercooling, leading to the formation and growth of eutectic microconstituent. The alterations of the eutectic morphology observed in this system are also attributed to recalescence phenomena: As soon as temperature rises above the T0-B2 point, the undercooling for eutectic growth is considerably high. High undercooling favors rod- or hive-like (commonly referred as fibrous) eutectic morphologies [47]. Taking into consideration the postulates of Li and Kuribayashi [47], according to which the eutectic constituent commences at the central areas of the remaining liquid, the hive-like eutectic morphology observed in the central areas of the remaining liquid observed in the present effort are, hence, expected. As the initial eutectic phases grow radially, recalescence causes further increase of the remaining liquid temperature, a fact that establishes conditions of lower growth kinetics and more intensive interphase diffusion, leading to more lamellar (regular) eutectic configurations.

#### 3.3.2. Alloy B: NiAl-30Cr-10Mo

It has been already mentioned in a previous paragraph that the overall microstructure complies with the predictions suggested by Peng et al. [24] according to which a primary phase and a eutectic constituent are expected to form. However, a detailed examination, especially of the primary phase, reveals great differences compared to the analogous primary phase in the previous alloy (Figure 5). EDS analysis (Table 1) indicates that the primary phase is rich in Cr and Mo with small amounts of Ni and Al being dissolved. It seems, therefore, that the substitution of 10% Cr by Mo stabilizes the A2 phase as the primary phase.

This A2 primary phase stabilization can be explained by the following phenomena that can act either individually of synergistically:(1)Based on EDS analysis (Table 1), the primary A2 phase contains Cr and Mo in relative ratio of Cr/Mo ≈ 70/30. According to the Cr-Mo phase diagram [48], this ratio leads to an increase of the liquidus temperature by several degrees compared to the melting point of Cr. This observation, in conjunction with the fact that Cr-Mo show complete solubility, leads to the fact that the presence of Mo controls the initiation of the solidification process, binds Cr form the liquid phase, and forms a primary modified A2 Cr phase rich in Mo. It is worth noticing, as well, that if the value of the melting temperature of the primary phase, as calculated by the parametric models (~2200 K), is taken into consideration, the tendency for the primary Cr(Mo) formation becomes even stronger.(2)The presence of Mo most likely affects the T0-A2 temperature presented by Tang et al. [15]. The presence of Mo, as stated previously, raises considerably the liquidus temperature. Having taken into consideration the fact that during arc melting, the temperature of heating and melting of the various systems was kept constant (arc parameters were not modified or changed), it is logical to assume that the increase of the liquidus temperature corresponds to a lower degree of liquid phase overheating, compared with the case of the NiAl-40Cr alloy. A lower degree of liquid phase overheating, as Mei and Li [49] and Yang et al. [50] postulate, leads to a reduction of the necessary for nucleation undercooling. In the case of the present work, this practically implies that the presence of Mo shifts the T0-A2 temperatures towards higher values, i.e., the necessary undercooling for primary Cr(Mo) A2 phase nucleation is reduced. It is strongly likely, that the established upon the initial stages of solidification, undercooling conditions were dropped below the affected T0-A2 Cr(Mo) values, and as such, the formation of A2 Cr(Mo) phase became dominant.(3)Synergistically to these potential mechanisms, another parameter that may also enhance the stability of the primary A2 Cr(Mo) phase is the lattice distortion value δ. As shown in Table 1, the calculated lattice distortion δ in the case of the primary NiAl B2 in alloy NiAl-40Cr is roughly 6.8, whereas in the case of the primary A2 Cr(Mo) in the present system, δ is roughly 4.8. This difference makes the A2 Cr(Mo) phase more favorable both in stability and kinetics of formation.

These potential phenomena can explain the formation of the core of A2 primary phase. However, as indicated in Figure 1b, a halo structure perimetrically to the basic Cr(Mo) core is observed. This type of perimetric phase formation around a primary phase in the case of alloy structures, very often, indicates the expression of a peritectic reaction. EDS analysis (Figure 5 and Table 1) indicates a sharp increase of Al and Ni content within the halo areas. A thorough investigation of the binary phase diagrams of the elements related to this system, surprisingly, revealed the presence of peritectic reaction in the case of Mo-Ni phase diagram around 1640 K [51] and in the case of Mo-Al phase diagram, at roughly 2330 K [52]. For the Mo-Al peritectic reaction to occur, an Al dissolution within Mo, a 20 at.% Al concentration should be exceeded. The EDS analysis of the primary phase reveals a relative Mo/Al ratio of roughly 74/24. Under these circumstances, it is possible that a first peritectic reaction at these temperature ranges can occur, leading to the formation of a first zone of peritectic reaction product, that of the AlMo_3_ intermetallic phase. Further temperature decrease will drive the system to a second peritectic reaction to occur, that of the Mo-Ni system, forming a second zone of peritectic reaction product, that of the MoNi intermetallic phase.

It is proposed thus for the primary phase that after the initial formation of the basic Cr(Mo) core, a sequence two zones of peritectic reaction zones is following, leading to the formation of the characteristic halo configuration peripherically to the basic Cr(Mo) core.

The solidification process is concluded with the formation of the eutectic microconstituent. EDS mapping analysis (Table 1) on the eutectic region shows a significant reduction of the Cr content compared to that of the primary phase (from 59% to 30 at.%). It also shows that the relative amounts of Ni, Al, Cr, and Mo correspond, with good approximation, to those of the eutectic composition according to Tang et al. [15], Duprin et al. [16], Peng et al. [24], and Demirtas et al. [53]. It is, thus, proposed that the formation of the primary phase consumes significant amount of Cr and Mo, leading the remaining liquid with such appropriate compositions for the formation of the final eutectic constituent. In terms of undercooling, it seems that the completion of the primary phase, raises the temperature of remaining liquid due to recalescence above T0-B2 and T0-A2, so that no decoupled growth is observed and regular eutectic structure is obtained. The development of the eutectic follows a sequence of a refined pattern at the central areas due to faster nucleation and growth kinetics and a radial development of coarser morphologies due to recalescence and heat release as the initial eutectic phases grow.

#### 3.3.3. Alloy C: NiAl-25Cr-15Mo

The general view of this alloy system, as presented in Figure 1, does not seem to alter significantly from that of the previous alloy, i.e., a primary phase with the characteristic perimetric halo followed by a eutectic structure can be observed, Figure 6. A slight difference compared to the previous alloy is the fact that the eutectic microconstituent appears to be slightly restricted. This observation is in agreement with the predictions of Peng at al. [24] where by increasing the Mo content, the system diverges from the eutectic liquid surface and the extent of the eutectic region is expected to be reduced compared to the previous alloy.

As far as the primary phase is concerned, no significant alterations of its formation sequence and mechanism can be noticed. Mo content in the primary phase core is raised (EDS analysis in Table 1) with the relative Cr/Mo being 47/53, making Mo the dominant factor to drive the solidification events. This practically implies a sequence of further increase of the liquidus temperatures, further lowering of the necessary for nucleation undercooling, and further increase of the modified T0 – A2 Cr(Mo) temperature. All these points, in conjunction with the constant low value of lattice distortion δ for the primary phase (4.87), eliminate the possibility of primary B2 phase to form and so is the case. Concerning the perimetric halo zone, as in the previous case, EDS analysis (Table 1) shows a significant increase of Al and Ni within from the core towards the perimetric zone. The relative ratio of Mo/Al is 82/18. This ratio, although below the 80/20 ratio predicted by the Mo-Al phase diagram [52] for the peritectic reaction to occur, it is, however, very close to this limit and in conjunction with the presence of other alloying elements being present and the equilibrium conditions experienced upon manufacturing, it is possible that the peritectic reaction does occur leading, as in the case of the previous alloy, to the formation of the first zone of peritectic product (AlMo_3_ intermetallic phase). Once the first peritectic zone is formed, further temperature reduction leads to the second peritectic reaction and the formation of the second peritectic zone product, that of the MoNi intermetallic.

Following the primary phase formation, further temperature decrease leads to the onset and development of the eutectic phases. It seems, however, that compared to the previous case, the eutectic areas are suppressed with parallel expansion of the dark phase within the lastly to solidify regions (Figure 6). This observation is a first hint that after the formation of the primary phase; the remaining liquid does not behave like a solid eutectic composition system, but rather like a hypoeutectic system. Indeed, as EDS mapping analysis within the interprimary phase regions showed that the Cr content was reduced form ~31 at.% in the previous alloy to ~21 at.% in the present system. Since, the interprimary area was almost entirely a eutectic phase in the previous alloy, it is logical to postulate that such dramatic Cr reduction shifts the compositional conditions of the remaining liquid towards hypoeutectic concentrations. As such, the solidification of the interprimary liquid follows a path of primary A2 NiAl and a final stage of eutectic NiAl-Cr formation.

#### 3.3.4. Alloy D:NiAl-20Cr-20Mo

Further increase of Mo continues the tendency already observed for the two previous alloys, i.e., further reduction of the eutectic constituent, Figure 7. The general microstructure (Figure 1) follows the predictions of Peng et al. [24], i.e., the composition shifts further off the eutectic liquidus surface, and as such, the eutectic microconstituent should be even more reduced and so is the case.

The characteristics of the primary phase do not seem to alter significantly compared to the two previous alloys. The relative ratio of Cr/Mo is approximately 28/72 (Table 1), which practically means that Mo more intensively controls the initiation of the solidification process. As in the previous two cases, the further increase of liquidus temperature, the subsequent undercooling decrease, and increase of T0-B2 temperature along with the persistently stable value of the lattice distortion (δ ~ 4.24) eliminate the possibility of the formation of primary B2 NiAl phase. As far as the peritectic halo perimetrically to the primary core, no increase of the Al content from the core to the periphery is observed (Table 1) and the relative amounts of Mo/Al in the primary core are 89/11. This ratio is significantly lower than the minimum 80/20 that could lead to the expression of the first peritectic reaction, and as such, a first zone of AlMo_3_ phase, as in the previous cases, is highly unlikely to have been formed. On the contrary, the second peritectic reaction proceeds regularly, and as such, the halo area consists exclusively by the MoNi intermetallic phase.

What is really interesting though, about the microstructure of this alloy, is the further modification of the last to solidify liquid area. It is profound that the eutectic constituent is almost vanished. Following the approach of the previous alloys, EDS mapping in the area of the remaining liquid shows a further reduction of Cr to ~17.5 at.%. This practically means that the remaining liquid composition shifts even more towards the hypoeutectic region, and as such, its solidification comprises a vast first stage of primary B2 NiAl formation, followed by a limited eutectic microconstituent. However, this Cr reduction is highly unlike to cause such a dramatic reduction of the eutectic phase by itself. There must be also a strong effect on the T0-B2 temperature that will affect the overall solidification procedure. The authors suspect that the presence of Mo increases both the liquidus temperatures of both sides of the Tang et al. [15] NiAl-Cr diagram. As such, the necessary for primary nucleation undercooling in the hypoeutectic area is also reduced, as in the case of the hypereutectic alloy. The T0-B2 has increased at such point that the experienced undercooling ensures the formation of primary NiAl phase, yet the resulting recalescence increases the remaining liquid temperature to an extent very close to the eutectic phase reaction, providing a narrow marginal temperature undercooling for the eutectic sequence. In this way, the eutectic microconstituent can be severely restricted, as observed in the present case. Alternatively or synergistically to these effects, another possible reason for this eutectic phase restriction could be the fact that because of the off-equilibrium solidification conditions, the maximum Cr dissolution in the NiAl-Cr phase diagram may be drifted to higher than the equilibrium values, and as such, the composition may fall within the area where no eutectic microconstituent can be developed. The authors, nevertheless, do admit that further experimentation is demanded in order to clarify this solidification behavior.

#### 3.3.5. Alloy E: NiAl-15 Cr-25 Mo

The tendency observed in the previous cases that the increase of Mo suppresses the eutectic morphology, as predicted by Peng et al. [24], it seems to be followed in the present system as well. It is profound (Figure 8) that the eutectic microconstituent is almost vanished.

The primary phase morphology and characteristics do not seem to differ significantly from the previous case. A relative ratio Cr/Mo within the core of the primary phase has a value close to 24/76, and as in the previous case, Mo drives the initiation of the solidification sequence and determines the characteristics of the primary phase. Lattice distortion persists to low values (~4.04) and in conjunction with the further increase of the liquids’ temperature and the T0-A2 temperature, ensures the stability and the domination of the A2 phase. As far as the halo phase is concerned, the Mo/Al ratio in the center of the core is roughly 90/10, and as such, the formation of a primary MoAl peritectic reaction product is not expected. Indeed, EDS analysis (Table 1) did not show any increase of Al in the halo areas, and as such, MoAl is unlikely to form. As in the previous case, a single peritectic zone of MoNi phase is established.

Questions are arisen, yet, as far as the solidification of the last liquid is concerned. EDS mapping analysis of this area revealed a further decrease of the Cr content close to 15 at.%. The obtained microstructure suggests that the depletion of Cr by the primary phase shifts the composition of the remaining liquid even deeper to the hypoeutectic area, and following the behavior Mo may have on the liquidus and the T0-B2 temperatures, it is most likely that the experienced undercooling provided the conditions for primary B2 phase formation, yet recalescence and heat release did not provide extended undercooling conditions for eutectic constituent formation. As such, the eutectic area is further suppressed.

#### 3.3.6. Alloy F: NiAl-10Cr-30 Mo

The overall microstructure is in compliance with the predictions of Peng et al. [24]. In-depth examination reveals that the morphologies do not differ significantly with the previous case; with the main conclusion being the extinguishing of the eutectic phases, as expected, for reasons described previously.

Another important observation (Figure 9), however, that has to be mentioned is the restriction of the peritectic halo zone. A possible explanation for this phenomenon is the gradual increase of Mo from alloys (b) to (f) with a subsequent decrease of the relative Ni/Mo atomic ratio within the core of the primary phases. Indeed, based on EDS analysis (Table 1), this ratio starts with a value of Ni/Mo ~22/78 in the case of alloy (b) and falls down to a value of roughly 6/94 in the case of the present alloy (alloy f). According to the binary Mo–Ni phase diagram, the maximum solubility on Ni into Mo does not exceed a value of 2 at.%. This practically means that even at the case of minimum Ni/Mo ratio (6 at.% in the present alloy), the peritectic reaction is prone to take place, yet by simple level rule considerations, the extent of the peritectic reaction product (MoNi intermetallic) is expected to be reduced, as is the actual case, observed by the microstructural features. Apart from this slight difference, no other severe microstructural modification can be distinguished.

#### 3.3.7. Alloy G: NiAl-40Mo

This final alloy contains no Cr, and as such, its microstructural features should be approached by the pseudobinary NiAl-Mo phase diagram, as proposed by Peng et al. [24]. Based on this phase diagram, there is a eutectic point of Mo composition roughly at 10 at.% and at temperature close to 1900 K. Since the nominal composition of this alloy is 40 at.% Mo, the system is a hypereutectic one, and as such, its microstructure should consist of primary Mo phase and a eutectic NiAl-Mo mixture. Surprisingly, as Figure 10 shows, the microstructure consists of a light primary phase, a dark phase, and a eutectic mixture. It is also interesting to notice that no halo zone peripherally to the primary light phase can be observed. EDS analysis (Table 1) shows that the light phase consists almost entirely of Mo, whereas the dark phase consists almost exclusively of Ni and Al. The eutectic microconstituent is a mixture of NiAl and Mo. The observed microstructure strongly indicates that the pseudobinary NiAl-Mo phase diagram proposed by Peng et al. [24] is not adequate by itself to explain the solidification sequence. Based on the work of Tang et al. [15], however, it is profound that other parameters such as T0 temperatures, undercooling, and recalescence should account for the finally received microstructure. To the best of the authors knowledge, no research work is reported in the international bibliography calculating and commenting on the T0-A2 (Mo) temperature. As such, taking into consideration: a) the behavior of the NiAl-Cr system proposed of Tang et al. [15] with the described T0-B2 (NiAl) temperature, b) the fact that Peng et al. [24] proposed a similar pseudobinary NiAl–Mo system, and c) the perfect solubility between Mo-Cr according to the binary Mo-Cr phase diagram, the authors suspect that a correspondence T0-A2 (Mo) temperature should behave in a similar manner as that of T0-A2 (Cr) phase.

Under this phenomenologically logical assumption, the solidification sequence can be as follows:(1)Upon solidification initiation, undercooling falls below the T0-A2 (Mo) and T0-B2 (NiAl) temperatures establishing such conditions for decoupled, partitionless growth of Mo and NiAl phases, with Mo being the first one to form due to its higher melting point.(2)As soon as recalescence is expressed, temperature is raised so that Mo continues to grow in a partitioning to the liquid mode, whereas NiAl proceeds in a partitionless way.(3)Further increase of temperature in the remaining liquid establishes conditions for partitioning growth of both NiAl and Mo establishing conditions for the development of the eutectic mixture growth, as the final stage of the solidification process.

Another important observation is the lack of halo zones at the periphery of the light phase, which is an indication of significant suppression of the peritectic reaction product. Indeed, as EDS analysis indicates (Table 1), the relative Ni/Mo ratio in the core of the light phase has reduced even further compared to the previous alloys, to values roughly 3/97 and as such the peritectic reaction extent is further restricted to point that cannot be observed in the final microstructure.

### 3.4. Microhardness and Wear Response

Figure 11 presents the microhardness values and the wear rates of the different systems produced in the present effort. As far as the microhardness is concerned, it can be noticed that there is a gradual increase until the case of alloy D (NiAl-20Cr-20Mo) followed by a gradual decrease in the case of alloy G (NiAl-40Mo). This tendency is in agreement with one of the fundamental characteristics of high-entropy alloys, the so-called cocktail effect, according to which at the maximum entropic content, the systems shows exceptionally increased mechanical properties, far beyond the properties of each individual element participating in the system [54,55,56]. This is indeed the tendency observed in the present case, as alloy D has the maximum entropic contribution effect (Table 2), i.e., the maximum expression of the cocktail effect enhancement. This is, however, a general observation, as far as the peak microhardness of alloy D is concerned.

Figure 12 shows the average mass loss vs. the sliding distance for each individual alloy. The first 200 m of the experiment were selected as a starting point for all the calculations due to the fact that, at the initial stages of the test (sliding distance between 0 and 200 m), data do not follow a linear progression when listed with the following ones. Such phenomenon could be associated with the reduction of the mechanical interactions between the examined materials and the counter body ball surface as sliding evolves over longer distances. More specifically, the high pressures, which develop at each contact point, lead to the spreading or penetration of individual asperities that stimulate an irreversible change in the shape of the contacting asperities and is thus nonreproducible [57]. However, it can be easily observed when comparing data in Figure 12 with wear rate values in Figure 11 that the same trend is followed. In particular, alloys with a lower degree of mass lost during the test exhibit lower wear rates and vice versa.

When focusing in Figure 11, an important observation is the fact that microhardness measurements and wear rates do not follow the classic trend proposed by Archard, according to which, the harder the material the lower the wear rate, i.e., the more increased its wear resistance. Due to this significant observation, the wear response of the various alloys should be approached and explained in terms of the microstructural and wear track morphological features. For these purposes, the authors suggest three discrete cases to explain the wear behavior:

#### 3.4.1. Case 1: the NiAl-Cr alloy

Figure 13 presents various magnifications of the wear track morphology of this alloy system along with the results of EDS point analysis on selective areas. It can be observed that the material removal areas are those associated with the primary A2 and B2 phases. It seems that the interfacial area between the coarse lamellae eutectic structure and the primary A2 and B2 primary phases are regions of weakness, where upon, the sliding and shear action cannot support the primary phase grains leading an extensive material removal leading to the calculated wear rate. EDS analysis showed the presence of oxide phases, of relatively increased Fe content, within the cavities of the detached primary grains. This oxide phases are most likely debris being formed upon testing with Fe being provided by the steel ball counter body. In general, and compared with the other systems, the wear rate is relatively low. The absence of the extensive grooves along the sliding direction suggest absence of abrasive wear, with the main mechanism being the delamination-detachment mode of the primary phases, as mentioned previously.

#### 3.4.2. Case 2: alloys with progressively increasing Mo content with Cr being present (alloys B to F)

The introduction of Mo, resulted in the expression of an obscure situation: despite the fact that the presence of Mo leads to an increase of microhardness, the wear resistance does not seem to follow the same trend. A close examination of the wear track morphologies of these systems (Figure 14, Figure 15, Figure 16, Figure 17 and Figure 18) reveals a series of interesting points, as follows:(1)A peak value of the wear rate is observed for alloy B, which is gradually reduced until the case of alloy F. This practically means that the introduction of small amount of Mo bursts the wear rate at very high values, yet the progressive increase of the Mo content leads to a gradual reduction of the wear rates.(2)All wear tracks (Figure 14, Figure 15, Figure 16, Figure 17 and Figure 18) show an extensive material loss at the areas of the lastly solidified liquid, irrespectively of this area’s morphological feature (being a eutectic microconstituent or not). This observation is very crucial as it suggests that the interfacial areas between the lastly to solidify liquid and the primary phases are weak and their rigidity is highly questionable.(3)The oxide phases being present within the cavities of the removed materials are also of great importance. EDS analysis (Figure 14, Figure 15, Figure 16, Figure 17 and Figure 18) on these selective areas shows that in the case of alloy B, the oxides phases (debris formed upon testing) are a mixture of Fe and Cr oxides. As the Mo content increases, the composition of these oxides is altered. In alloy C, the oxide phases are rich in Fe rather in Cr, and in the case of alloy D, the Mo content of the oxide phases is taking the lead. Further increase of Mo, alloy E, shows that the Ni is the dominant element followed by Fe. The Cr content is significantly reduced. Finally, in the case of alloy F, the Ni content is further increased and the Cr content is further reduced. As it will be discussed in a following paragraph, these elemental variations play a vital role on the lubricating action of the oxide phases being formed.(4)Even among these alloys, a classification of their wear rates can be conducted: alloys B and C forming the first group of high wear rates, alloys D and E forming a second group with reduced wear rates, and finally alloy F with considerably reduced wear rate. There must be a reason for this kind of transition which will be addressed later.(5)Another observation that will help in the explanation of the wear behavior of the different systems is the morphology of the primary phases and, especially, the morphology of the contour of the primary phases. By recalling the involved microstructures (Figure 5, Figure 6, Figure 7, Figure 8 and Figure 9), it can be observed that the periphery of the primary phases in the cases of alloys B to E is characterized by an intensive jigsaw like morphology with acicular protrusions towards the lastly to solidify liquid. Alloy G shows a more planar, yet smoother perimetric contour of the primary phase.

Based on these important observations, a possible explanation of this category of system wear response could be as follows:(1)The first crucial factor, as already mentioned, is the nature of the oxides formed. The observed oxides in these systems are Fe-based, Ni-based, Mo-based, and Cr-based oxides. According to the extensive reviews of Glascott et al. [58] and Stott and Wood [59], among these oxides, the most effective in performing lubricating action are the Fe-, Ni-, and Mo-based oxides which belong to the category of the so-called ductile oxides. Glascott et al. [58] postulates that these oxides under conditions of hydrostatic pressure show significant ductile behavior and can deform plastically. Glascott et al. [58] also mention that under sliding wear conditions, a high portion of hydrostatic pressure field (even as high as 60% of the overall stress field) especially at the areas of oxide asperity junctions, exists. Under this frame, the ductile oxides deform plastically and absorb a great amount of the shear stresses involved energy and provide to the system an effective lubrication action. Cr-based oxides on the other hand, despite the fact that they are harder, they are also very brittle and do not deform plastically as the other oxides, and as such, their lubrication action is limited. Stott et al. [59] also mentioned that another problem with the Cr-based oxides is their poor adhesion with the substrate especially in the case of Ni-Cr containing alloys. If we recall not the present experimental findings, it can be seen, in reverse order, in the alloys of high Mo content (alloys G, E, and D) that the oxides being formed are mainly Mo, Ni, and Fe-based, whereas the presence of Cr-based is very restricted. It is expected, thus, that in these systems, the oxides been present will perform a strong lubrication action. In the case of alloys C and B, it can be observed that the relative amounts of Cr are increased and as such, the lubricating action is reduced.(2)The severe mass loss, as already mentioned, is related to the extensive detachment of the last to solidify liquid areas. It should, hence, be a gradual weakening of the interfacial area between the primary phases and these lastly solidified regions. The difference between alloy A and alloys B to F, concerning these interfacial areas, is the presence of peritectic halos around the primary phases. As mentioned previously, the result of these peritectic reactions is the formation of strong and brittle intermetallic phases. Taking into consideration that as the Mo content increases from alloys B to C, the second primary phase to be formed is the NiAl intermetallic phase; Figure 19 shows a graphical representation of the possible epitaxial mismatches between the phases gradually formed from the core of the primary phases, through the formation of the peritectic halo intermetallics, to the last formation of the secondary primary NiAl. Alloys B and C belong to the first category of phase formation (Figure 19a) where both the peritectic intermetallics are present and the epitaxial mismatch starts from ~10%, reduced to ~8%, and finally, to ~2% at the outer layers. On the contrary, alloys D and E did not show the formation of the first peritectic halo reaction zone, and as such, the epitaxial mismatch (Figure 19b) starts with values as low as ~1% moving to ~2% at the outer layer. Intensive epitaxial mismatch can be a reason for intensive residual stresses being developed, and as such, the integrity and the stability of the interfacial areas are strongly and negatively affected [60]. This postulate is in agreement with the experimental findings of this effort. The interfacial rigidity in the case of alloys B and C is more affected due to the higher values of epitaxial mismatches, and as such, the danger of a detachment event is highly increased. On the contrary, the progression from the primary phase to the last solidified areas is smother in the case of alloys D and E, in terms of interfacial stability, and as such, the possibility of a forthcoming detachment event is reduced. Indeed, this is what is actually observed in the present case where alloys B and C showed significant material loss events in the last solidified regions compared to the same phenomenon in the case of alloys D and E. Furthermore, in the case of alloy F where the formation of peritectic halo is limited, the wear rate is further decreased.

The presence of intermetallic phases perimetrically to the primary grains, can also affect the integrity of the interfacial areas in other terms. As mentioned previously, the contour of the primary phases in alloys B to F has a strong acicular jigsaw morphology. This aggressive landscape can also establish a field of high residual stresses that may lead to a premature collapsing of the interfacial area. The acicular morphologies are also the result of the formation of intermetallic phases, which according to the classic nucleation theory shows increased values of the so-called Jackson factor (over the value of 2), which is an indication of a faceted growth of mode that, in turn, leads to the formation of such kind of acicular morphologies [61].

#### 3.4.3. Case 3: the NiAl-40Mo Alloy

Based on the remarks mentioned for the alloys in the previous case, in this alloy, the dominant oxides are Ni, Fe, and Mo (Figure 20), i.e., oxides of high lubricating action. Additionally, no significant evidence for the formation of intermetallic peritectic halos can be distinguished, which means that no epitaxial mismatch residual stress field is established that can weaken the interfacial area between the primary phases and the last to solidify liquid. Furthermore, the contour of the primary phases, exactly due to the absence of intermetallic phases, is smooth and rounded and as such no further negative effect on residual stress field building up by acicular morphologies takes place.

In summary, the wear response of the alloys produced on the present effort is a combination of the factors reported previously, i.e., the nature and chemistry of the surface oxides and the presence, the nature, and the morphological features of intermetallic phases. It has to mentioned, nevertheless, that examination of cross section areas related to the wear tracks, as adopted in other experimental efforts [62,63], would provide a further insight on the wear phenomena and such an assessment is currently being undertaken.

## 4. Concluding Remarks

### 4.1. Microstructural Features

(1)The microstructure of the NiAl-40Cr alloy is a result of solidification sequence according to which primary A2(Cr) and B2(NiAl) phases are developed partitionless due to undercooling phenomenon. Recalescence and its involved heat release lead to the development of a final eutectic structure.(2)The substitution of Cr by 10% Mo, in the case of alloy B, stabilizes the A2 phase as a primary phase as both Mo and undercooling do not favor a parallel B2 growth in a partitionless way. Phase diagram predictions verified in practice and multiple AlMo_3_ and MoNi intermetallic phases are developed at the periphery of the primary grains due to peritectic reactions. Recalescence leads to the final stage of eutectic phase formation.(3)By increasing the Mo content in all the preceding alloys, the A2 phase is further established as the primary phase, AlMo_3_ is eventually vanished, and the eutectic phase is continuously restricted. B2 (NiAl) is gradually increased as a secondary (following A2) “primary” phase. All these microstructural sequences are related to partitioning/partitionless growth, recalescence, and undercooling phenomenon.(4)The NiAl-40Mo alloy consists of primary Mo, NiAl, and NiAl-Mo eutectic constituents. The same phenomena, as in the previous cases, are responsible for this solidification sequence.(5)The increase of the Mo content leads to progressive elimination of the eutectic microconstituent and establishes significantly simpler microstructures of greater controlling potential.

### 4.2. Sliding Wear Response

(1)The sliding wear behavior of alloy NiAl-40Cr showed extensive material loss due to localized delamination of the eutectic areas. Oxides of Fe and Cr are the main oxides being present.(2)An increased wear rate was observed in the case of alloy B (NiAl-30Cr-10Mo) by extensive material loss at the eutectic areas, due to the presence of intermetallic peritectic halos and the nonlubricating action of the formed oxides. The increase of the Mo content leads to the reduction of the wear rate (from alloy B to alloy F), due to the gradual restriction of the intermetallic phases and the gradual formation of oxides phases with intensive lubrication action.(3)The NiAl-40Mo is characterized by a relatively low wear rate due to the absence of intermetallic phases and the presence of lubricating oxides.

## Figures and Tables

**Figure 1 materials-13-03445-f001:**
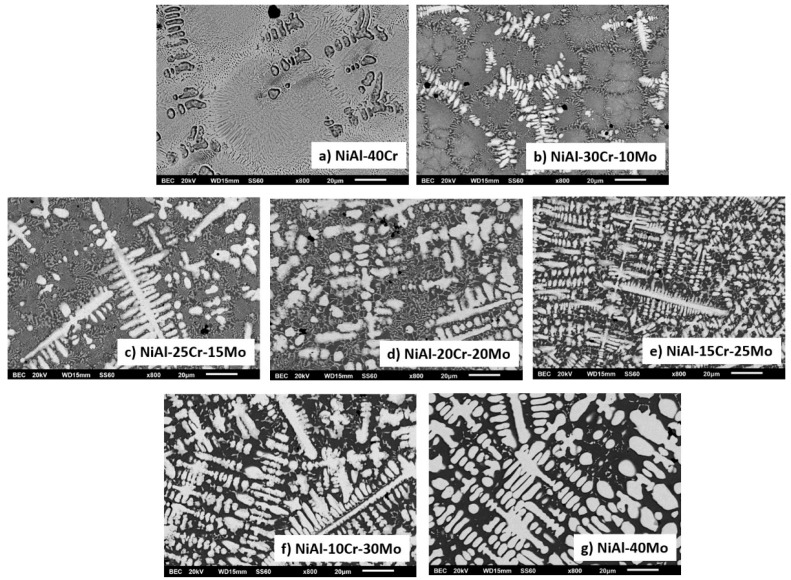
SEM images showing a panoramic view of the different systems produced in the present effort: (**a**) NiAl-40Cr alloy showing primary phases and eutectic areas of multiple morphologies, (**b**) NiAl-30Cr-10Mo alloy showing primary phases and eutectic microconstituent. Perimetric to the primary grains halo zones can be spotted, (**c**) NiAl-25Cr-15Mo alloy consisting of primary phase, halo zones, and eutectic phases, (**d**) NiAl-20Cr-20Mo alloy with a first primary phase surrounding by halo zones, traces of second primary phase, and eutectic microconstituent, (**e**) NiAl-15Cr-25Mo alloy showing a first primary phase with perimetric halo zones, a second primary phase, and a eutectic phase significantly restricted, (**f**) NiAl-10Cr-25Mo consisting mostly of two primary phases. The eutectic areas are difficult to be distinguished. (**g**) NiAl-40Mo consisting of two primary phases and limited eutectic areas.

**Figure 2 materials-13-03445-f002:**
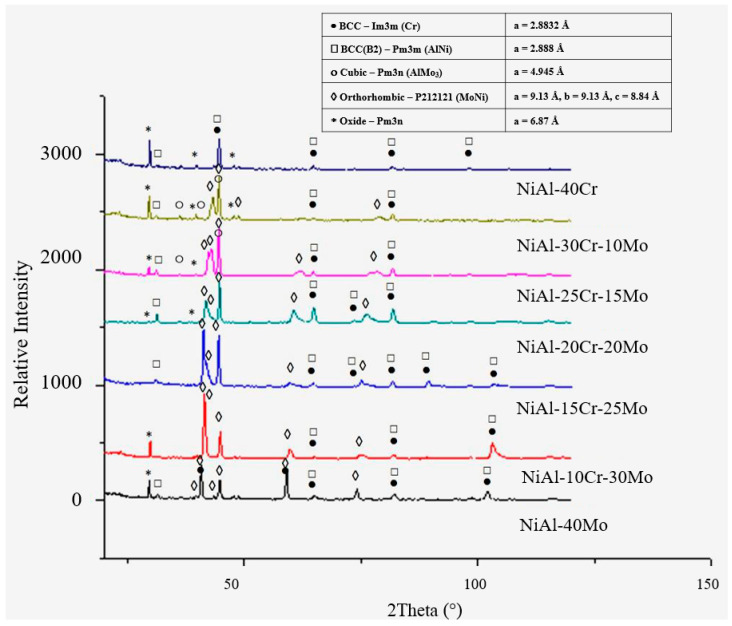
XRD graph showing the evolution of the phases with altering Mo content. Alloy A consists of two BCC phases, alloys B, C, and D also contain two BCC structures along with two intermetallic phases, alloy E and F contain two BCC structures and one intermetallic phase, and alloy G contain two BCC structures.

**Figure 3 materials-13-03445-f003:**
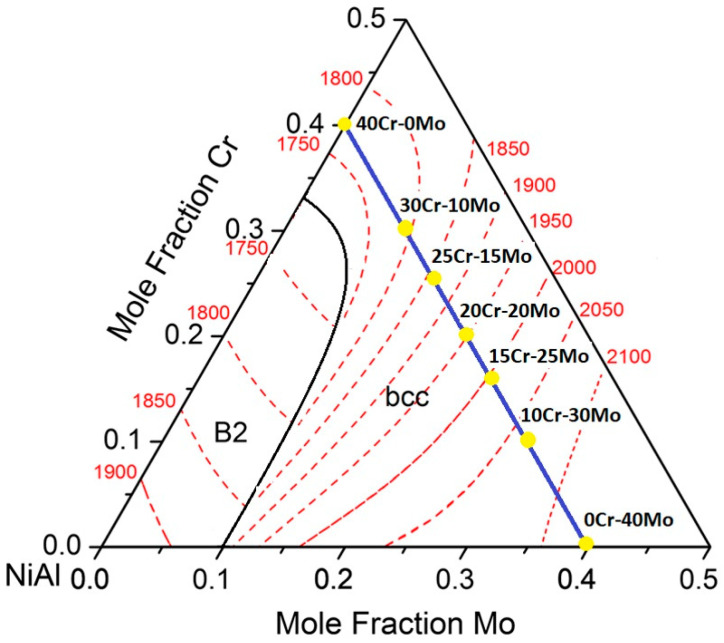
Graphical reconstruction of the ternary NiAl-Mo-Cr system as proposed by Peng et al. [24]. The blue line and the yellow spots indicate the composition of the system produced in the present effort.

**Figure 4 materials-13-03445-f004:**
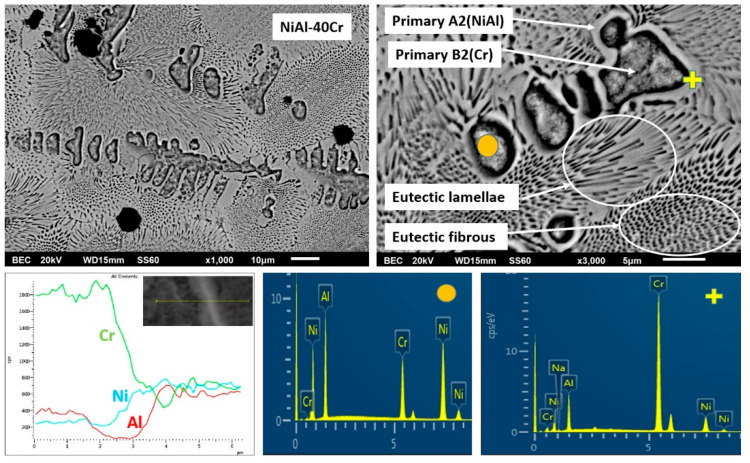
Higher magnification SEM images showing the microstructure of alloy A (NiAl-40Cr). Line scan EDS analysis and point EDS analysis are also included.

**Figure 5 materials-13-03445-f005:**
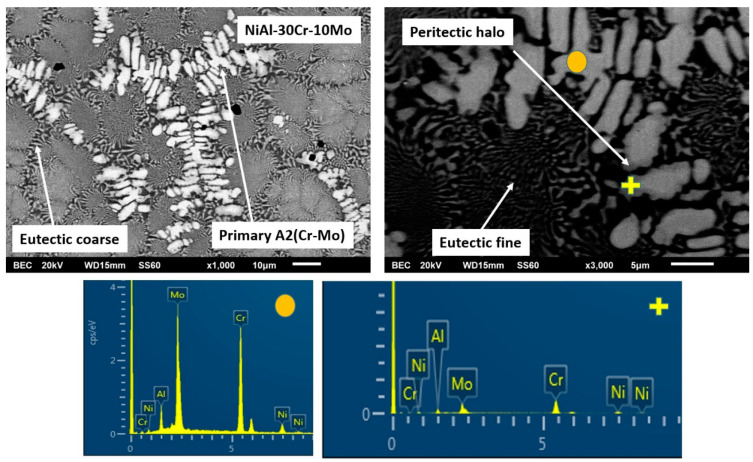
SEM images of magnification presenting the microstructural features of alloy B (NiAl-30Cr-10Mo). EDS point analysis on selective areas are also included.

**Figure 6 materials-13-03445-f006:**
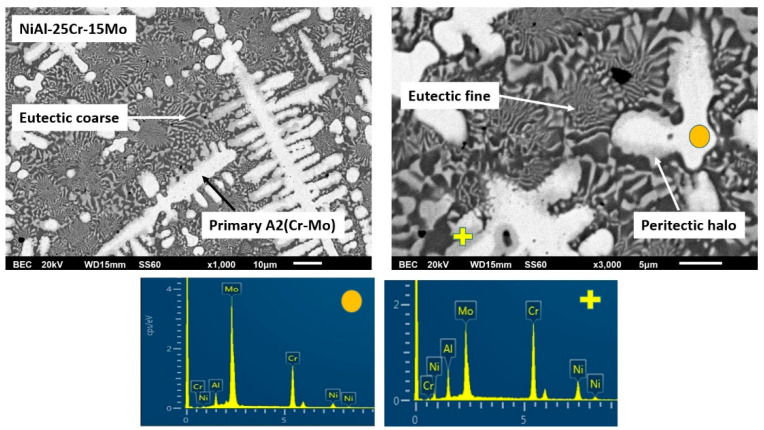
SEM images of higher magnification presenting the microstructural features of alloy C (NiAl-25Cr-15Mo). EDS point analysis on selective areas are also included.

**Figure 7 materials-13-03445-f007:**
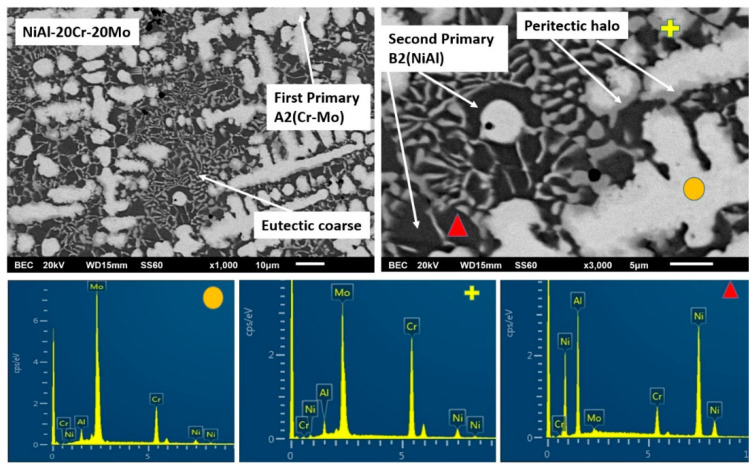
SEM images of higher magnification presenting the microstructural features of alloy D (NiAl-20Cr-20Mo). EDS point analysis on selective areas are also included.

**Figure 8 materials-13-03445-f008:**
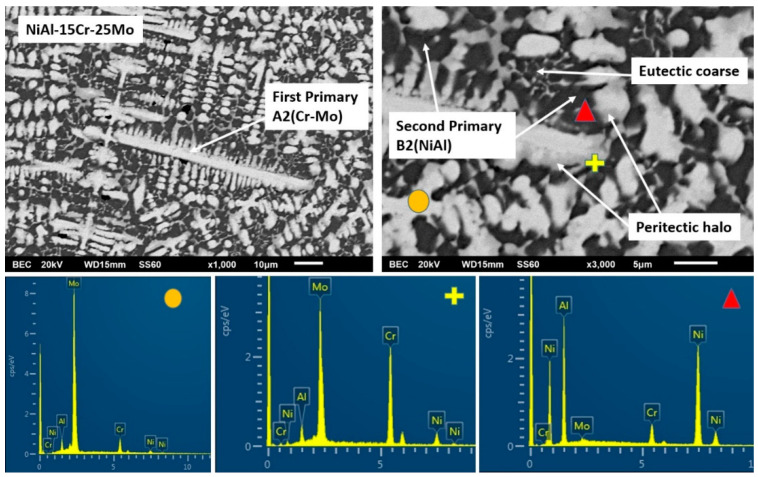
SEM images of higher magnification presenting the microstructural features of alloy E (NiAl-15Cr-25Mo). EDS point analysis on selective areas are also included.

**Figure 9 materials-13-03445-f009:**
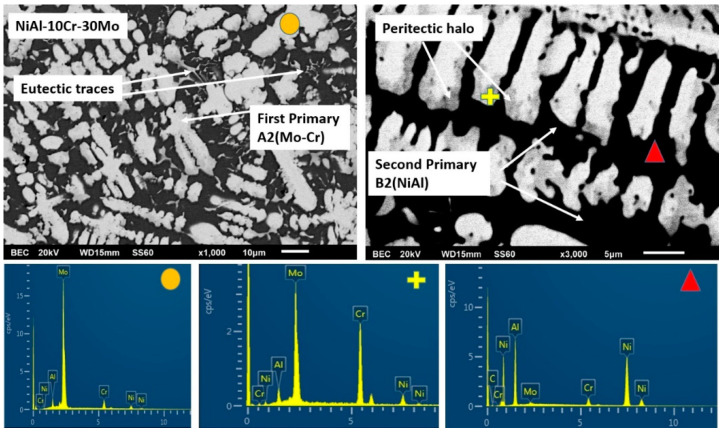
SEM images of higher magnification presenting the microstructural features of alloy F (NiAl-10Cr-30Mo). EDS point analysis on selective areas are also included.

**Figure 10 materials-13-03445-f010:**
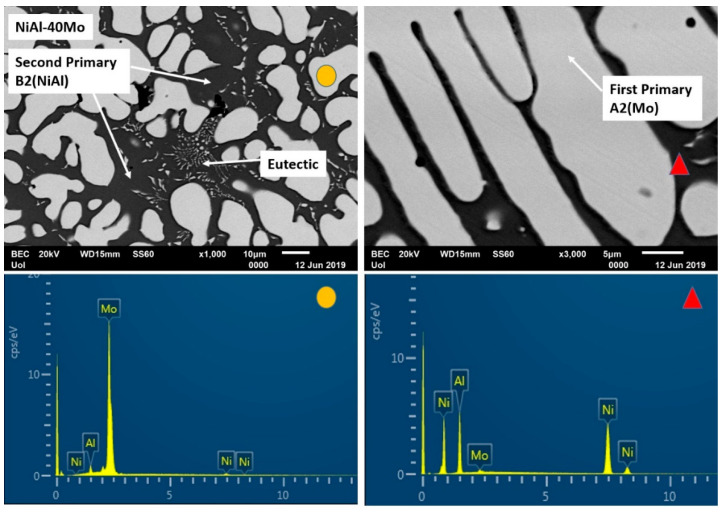
SEM images of higher magnification presenting the microstructural features of alloy G (NiAl-40Mo). EDS point analysis on selective areas are also included.

**Figure 11 materials-13-03445-f011:**
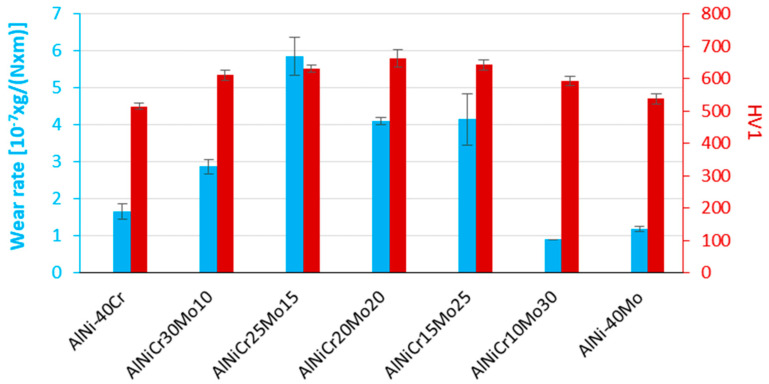
Microhardness values and wear rates of the different systems produced in the present effort.

**Figure 12 materials-13-03445-f012:**
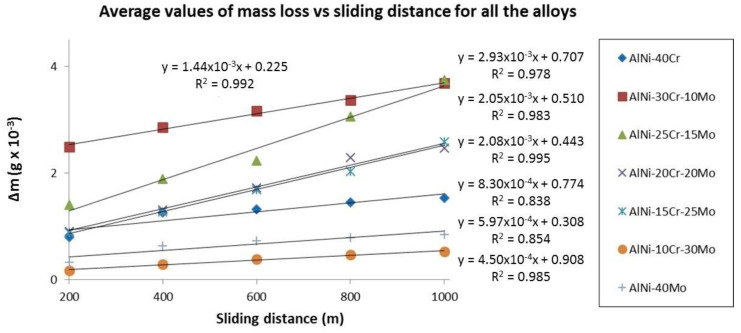
Average values of mass loss vs. sliding distance of the different systems produced in the present effort.

**Figure 13 materials-13-03445-f013:**
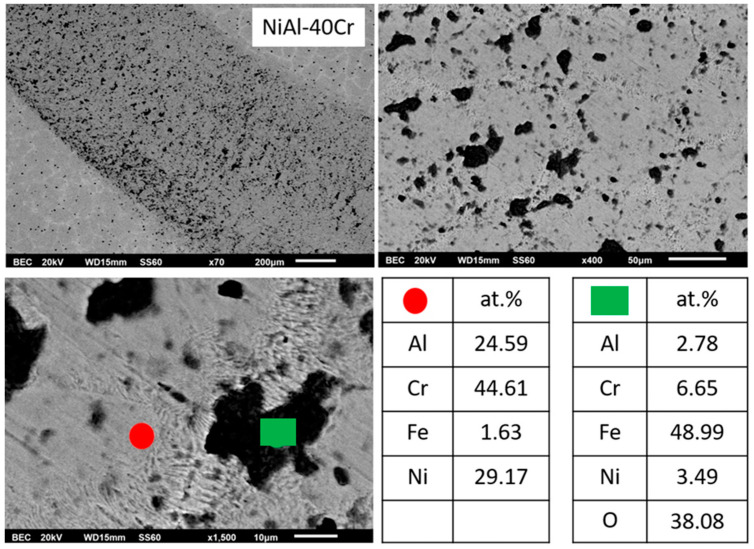
SEM images of different magnifications showing the characteristic features of the wear track morphology of alloy A (NiAl-40Cr). Tables of EDS point analysis measurements in selective areas are also included.

**Figure 14 materials-13-03445-f014:**
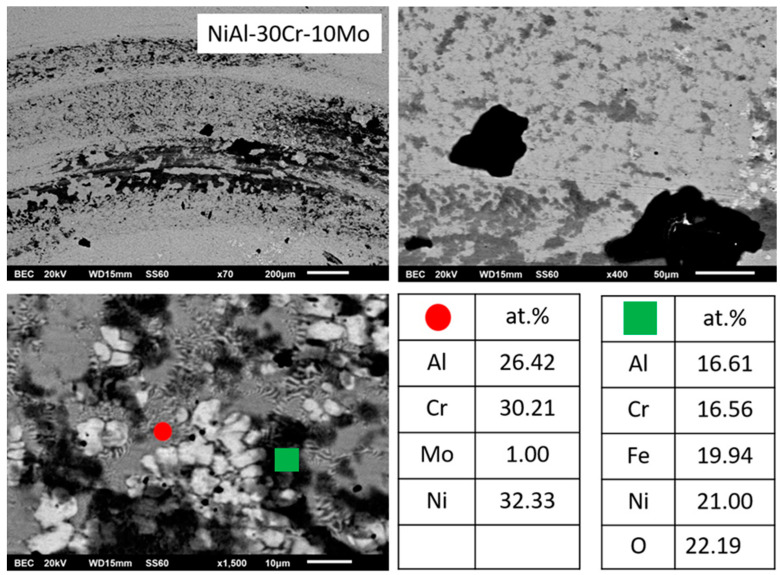
SEM images of different magnifications showing the characteristic features of the wear track morphology of alloy B (NiAl-30Cr-10Mo). Tables of EDS point analysis measurements in selective areas are also included.

**Figure 15 materials-13-03445-f015:**
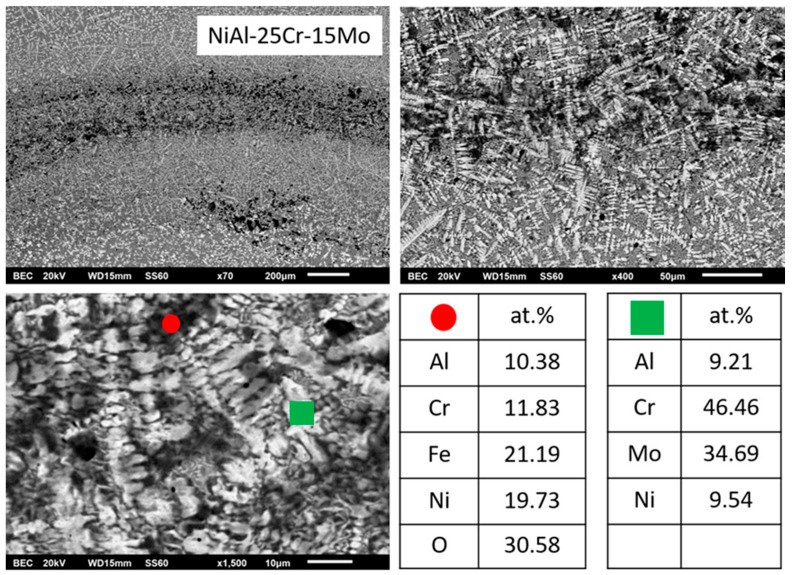
SEM images of different magnifications showing the characteristic features of the wear track morphology of alloy C (NiAl-25Cr-15Mo). Tables of EDS point analysis measurements in selective areas are also included.

**Figure 16 materials-13-03445-f016:**
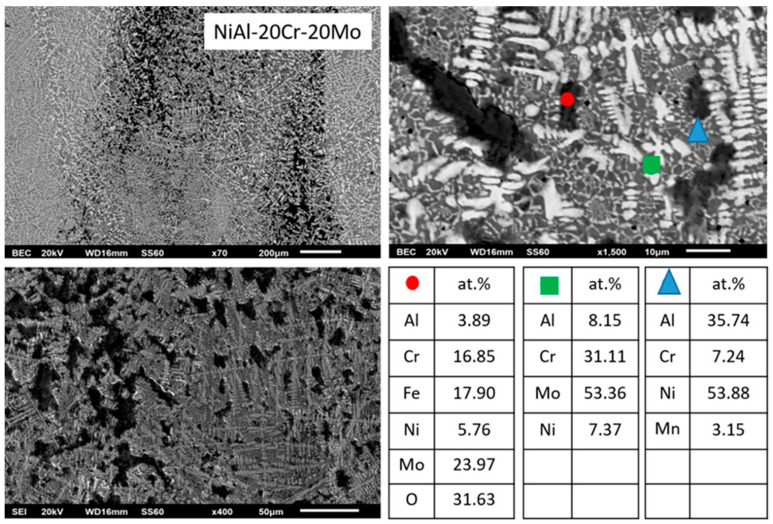
SEM images of different magnifications showing the characteristic features of the wear track morphology of alloy D (NiAl-20Cr-20Mo). Tables of EDS point analysis measurements in selective areas are also included.

**Figure 17 materials-13-03445-f017:**
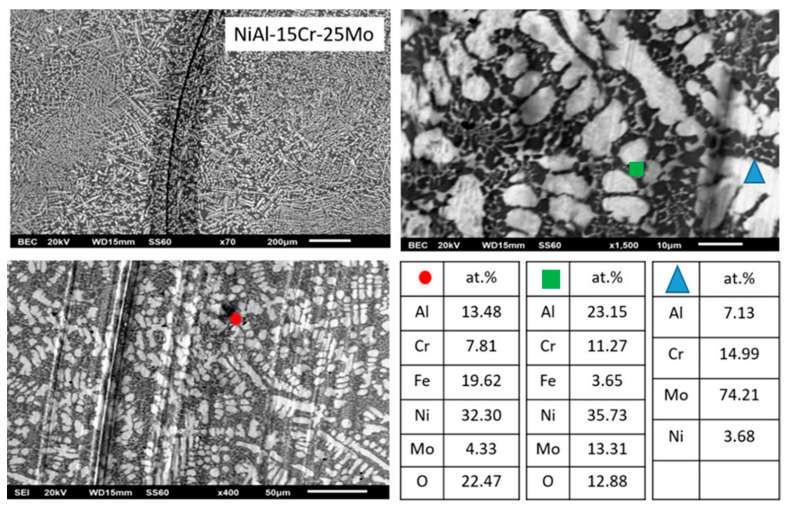
SEM images of different magnifications showing the characteristic features of the wear track morphology of alloy E (NiAl-15Cr-25Mo). Tables of EDS point analysis measurements in selective areas are also included.

**Figure 18 materials-13-03445-f018:**
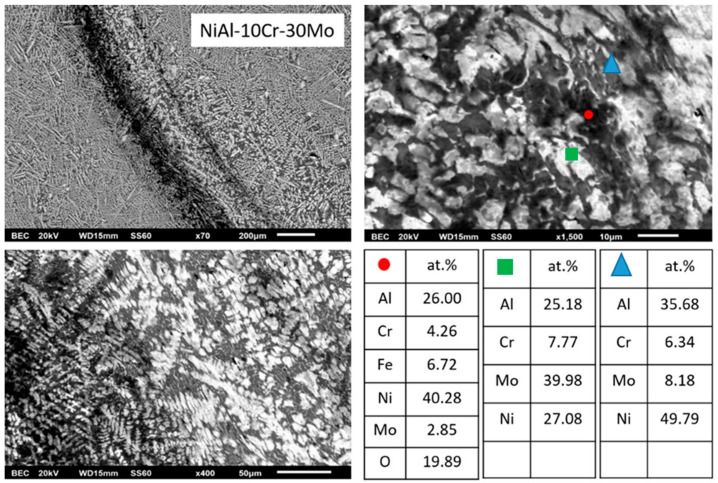
SEM images of different magnifications showing the characteristic features of the wear track morphology of alloy F (NiAl-10Cr-30Mo). Tables of EDS point analysis measurements in selective areas are also included.

**Figure 19 materials-13-03445-f019:**
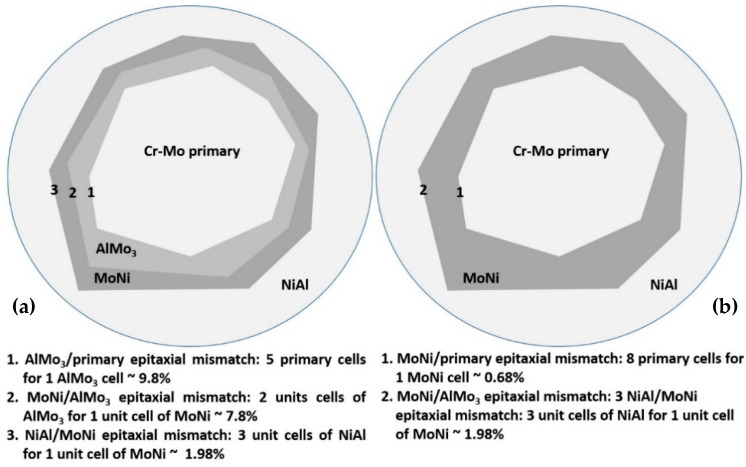
A schematic representation of the epitaxial mismatch evolution due to the presence of intermetallic phases. (**a**) sketch is related to the presence of both the peritectic intermetallic phases (AlMo_3_ and MoNi) and refers to alloys B, C, and D. (**b**) sketch is related to the presence of single peritectic intermetallic phase (MoNi) and refers to alloys E and F. Notice that the calculated values are the minimum ones, in both cases. Calculations were based on crystal lattice data provided by the PDF phase cards during the XRD analysis.

**Figure 20 materials-13-03445-f020:**
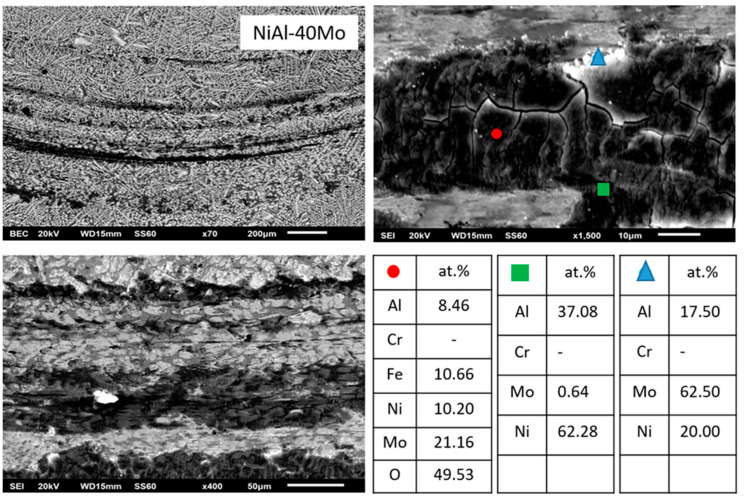
SEM images of different magnifications showing the characteristic features of the wear track morphology of alloy A (NiAl-40Mo). Tables of EDS point analysis measurements in selective areas are also included.

**Table 1 materials-13-03445-t001:** Nominal, actual, and individual phases compositions after EDS point and mapping analysis. Elemental ratios of interest are also included.

System	Composition (at.%)	Al	Ni	Cr	Mo	Mo/Cr + Mo	Mo/Ni + Mo	Mo/Al + Mo	Cr/NiAl + Cr	δ
Alloy A: AlNi-40Cr	Nominal	30	30	40	0	0	0	0	57/100	4.96
	Actual	26.21	33	40.76	0	0	0	0	0	
	Primary Dark	11.88	10.70	77.42	0	0	0	0	87.5/100	
	Primary Light	35.02	46.34	18.65	0	0	0	0	32/100	
	Eutectic Overall	36.92	43.76	19.32	0	0	0	0	32/100	
Alloy B: AlNi-30Cr-10Mo	Nominal	30	30	30	10	25/100	25/100	25/100	50/100	4.83
	Actual	25.7	33.2	30.99	10.11	25.5/100	23/100	28/100	52/100	
	Primary Light	8.09	7.01	58.98	25.91	30.5/100	78/100	76/100	88/100	
	Primary Halo	14.77	12.9	53.26	19.07	24/100	60/100	56/100	80/100	
	Eutectic Overall	39.47	47.83	10.77	1.92	15/100	4/100	4.5/100	20/100	
Alloy C: AlNi-25Cr-15Mo	Nominal	30	30	25	15	37.5/100	33/100	33/100	45/100	4.87
	Actual	23.74	33.96	25.99	16.31	39/100	32/100	41/100	47/100	
	Primary Light	9.54	6.47	39.48	44.52	53/100	87/100	82/100	83/100	
	Primary Halo	14.72	19.38	45.05	20.85	32/100	52/100	58/100	73/100	
	Eutectic Overall	29.26	39.06	21.66	10.01	32/100	20/100	26/100	39/100	
	Dark Phase	37.01	52.16	9.40	1.42	13/100	3/100	4/100	18/100	
Alloy D: AlNi-20Cr-20Mo	Nominal	30	30	20	20	50/100	40/100	40/100	40/100	4.33
	Actual	23.88	35.10	19.43	21.59	53/100	38/100	47.5/100	40/100	
	Primary Light	8.04	4.83	24.09	63.04	72/100	93/100	89/100	75/100	
	Primary Halo	6.13	7.93	54.28	31.65	37/100	20/100	16/100	88/100	
	Eutectic Overall	29.03	41.11	17.36	12.50	42/100	23/100	30/100	33/100	
	Dark Phase	37.51	54.84	6.98	0.67	9/100	1/100	2/100	13/100	
Alloy E: AlNi-15Cr-25Mo	Nominal	30	30	15	25	62.5/100	46/100	46/100	33/100	4.04
	Actual	23.44	34.70	14.82	27.03	65/100	44/100	54/100	34/100	
	Primary Light	7.3	3.58	21.91	67.11	75/100	95/100	90/100	80/100	
	Primary Halo	5.20	7.92	53.40	33.48	39/100	81/100	87/100	86/100	
	Eutectic Overall	28.77	41.58	15.82	13.83	47/100	25/100	33/100	31/100	
	Dark Phase	39.53	53.70	6.07	0.70	10/100	1/100	2/100	12/100	
Alloy F: AlNi-10Cr-30Mo	Nominal	30	30	10	30	75/100	50/100	50/100	25/100	3.63
	Actual	23.71	35.51	10.06	30.73	75/100	46/100	56/100	25/100	
	Primary Light	6.32	3.45	15.45	74.79	83/100	94/100	92/100	76/100	
	Peritectic Halo	Difficult to Ascertain								
	Eutectic Overall	33.99	47.46	7.33	11.82	62/100	20/100	26/100	15/100	
	Dark Phase	41.59	53.94	3.86	0.62	14/100	1/100	1.5/100	7.5/100	
Alloy G: AlNi-40Mo	Nominal	30	30	0	40	100	57/100	57/100	0	1.96
	Actual	23.06	37.30	0	39.65	100	52/100	63/100	-	
	Primary Light	6.91	3.11	0	89.98	100	97/100	93/100	-	
	Eutectic Overall	35.62	55.51	0	8.87	-	14/100	20/100	16/100	
	Dark Phase	41.13	57.94	0	0.96	100	1.5/100	2/100	2/100	

**Table 2 materials-13-03445-t002:** The values of the important parameters for the parametric models used in order to predict the formation of not of single solid solution phases.

System–Composition (at.%)	Atomic Percentage [at.%]	Zhang/Guo Criteria	King Criteria	Senkov Criteria
Al	Ni	Cr	Mo	δ	ΔS_mix_ [J/K·mol]	ΔH_mix_ [kJ/mol]	ΔH_f_ [kJ/mol]	ΔG [kJ/mol]	VEC	Ω	γ	Φ	Τ_m_ [K]	ΔH_IM_/ΔH_mix_	k_1_^cr^
NiAl-40Cr	Nominal	30	30	40	0	6.33	9.05	−16.08	16.57	−33.11	6.30	0.93	1.158	0.70	1650	1.95	1.19
Actual	26.21	33	40.79	0	6.10	9.00	−15.66	16.84	−32.46	6.53	0.97	1.158	0.68	1683	1.91	1.19
NiAl-30Cr-10Mo	Nominal	30	30	30	10	6.23	10.92	−15.48	18.12	−34.49	6.30	1.22	1.157	0.35	1726	2.07	1.24
Actual	25.7	33.2	30.99	10.11	6.05	10.89	−15.03	18.45	−33.84	6.56	1.28	1.158	0.35	1766	2.01	1.26
NiAl-25Cr-15Mo	Nominal	30	30	25	15	6.14	11.25	−15.18	18.90	−33.73	6.30	1.31	1.157	0.35	1765	2.14	1.26
Actual	23.74	33.96	25.99	16.31	5.90	11.25	−14.36	19.58	−33.07	6.65	1.44	1.157	0.35	1834	2.04	1.29
NiAl-20Cr-20Mo	Nominal	30	30	20	20	6.02	11.35	−14.88	19.67	−33.58	6.30	1.37	1.157	0.35	1803	2.22	1.27
Actual	23.88	35.1	19.43	21.59	5.83	11.29	−14.29	20.35	−32.94	6.69	1.47	1.157	0.35	1868	2.15	1.29
NiAl-15Cr-25Mo	Nominal	30	30	15	25	5.87	11.25	−14.58	20.45	−33.68	6.30	1.19	1.156	0.35	1841	2.30	1.28
Actual	23.44	34.7	14.82	27.03	5.71	11.17	−13.88	21.25	−32.91	6.68	1.54	1.157	0.35	1916	2.22	1.31
NiAl-10Cr-30Mo	Nominal	30	30	10	30	5.69	10.92	−14.28	21.22	−34.40	6.30	1.44	1.156	0.35	1880	2.40	1.29
Actual	23.71	35.51	10.06	30.73	5.62	10.82	−13.88	21.77	−33.30	6.71	1.51	1.157	0.35	1938	2.32	1.30
NiAl-40Mo	Nominal	30	30	0	40	5.22	9.05	−13.68	22.77	−32.44	6.30	1.29	1.55	0.68	1956	2.63	1.26
Actual	23.06	37.3	0	39.65	5.31	8.92	−13.54	23.16	−30.92	6.80	1.32	1.156	0.66	2007	2.52	1.26
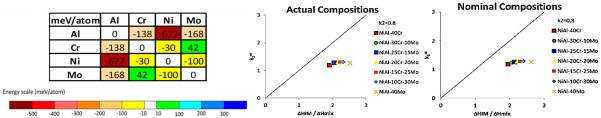

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
