# Peer review of "NiAl-Cr-Mo Medium Entropy Alloys: Microstructural Verification, Solidification Considerations, and Sliding Wear Response"

_materials, 2020, doi:10.3390/ma13163445_

Round 1

Reviewer 1 Report

In the reviewed manuscript the series of NiAl-Cr-Mo systems were produced and assessed as far as their microstructure and their sliding wear resistance. Results presented within the paper are of interest to steel community. The paper can be published after some minor issues have been addressed:

  1. Table 2 presents the values of the important parameters for the parametric models used in the presented work. The entropy of mixing between the alloying elements values calculated based on the Zhang/Guo criterion for different materials are marked of green and red colour. This mean that the value of the parameters calculated for NiAl-40Cr, NiAl-25Cr-15Mo, NiAl-10Cr-30Mo and NiAl-40Mo according to the limits presented in the text of the manuscript (line 745) was out of the acceptable values?
  2. The plots presented parameters for the nominal and actual compositions presented in the bottom part of Table 2 are unreadable. Axis description are hard to read.
  3. In references part can be found some edition mistakes. Please carefully proper the references part.

Reviewer 2 Report

The research paper is a well-structured and well-presented study. The introduction provides useful information about the relevance ofthe topic.

The method and the circumstances of the experiments are properly demonstrated, therefore the repeatibility of the experiments is ensured.

The analytic part of the study is also comprehensive; the applied figures and tables facilitate the easy-to-understand feature of the study.

The results are well-founded by the analysis.

The English of the text is required to be revised; a few errors can be found in the text. Revision by a native English-speaker is recommended.

Reviewer 3 Report

Paper is addressing Ni-Al-Cr-Mo system as a HEA alloy. It is interesting with a lot of experiments performed. However, some improvements are suggested listed below:

  1. English language needs proper polish by native speaker. There is bunch of misspelled words and errors. For example in line 12 "0Cr-40Mo". Some abbreviations are used before explanation, like EDS and SEM in lines 100 and 101, respectively.
  2. Experimental methods need better description. Was only 5 g of raw material melted and investigated? It seems very small amount of material for investigations. Were any heat treatments performed after material was made? For XRD analysis, no 2\theta range and dwell time is given. Was radiation monochromatic or not and did you use slits or not? Please state weight used for Vickers hardness.
  3. Figure 2 is missing.
  4. XRD is very crude. Was intendent only for phase analysis or quantitative analysis too? In the line for NoAl-40Mo it is very strange you detect Cr oxide if no Cr is present in the alloy. Also axis with values for intensity in stacked XRD does not have any meaning and values should be omitted. Some crude quantitative analysis could be done also from micrographs.
  5. Prior to experimental work usually initial predictions are made. It is my suggestion to change order of subsections.
  6. In Ni-Cr-Al alloy system it is common to form Ni3Al phase. Can you check if you have it or only B2 phase is present?
  7. EDS spots in Figures 12-17 are all marked with dots which are not clear.
  8. As you weighted discs every 200 m why not show this results in table or graph?

Reviewer 4 Report

The article deals with the preparation and characterization of medium-entropy Ni-Al-Cr-Mo alloys. It is an extensive study of the microstructure and properties of selected types of alloys.

Since the work in its discussion assesses the influence of chemical composition on the phase composition, I see the main deficiency of the work in the unconvincing phase analysis shown in Figure 2. This is a very low figure quality and does not prove the phase composition, which is the subject of further discussion. I therefore propose to perform a high quality X-ray diffraction measurement and analysis with refinement of microstructural parameters by the Rietveld method. Only then will discussion about the effect of chemical composition on phase composition relevant become relevant.

Quality of some figures needs to be improoved.

Reviewer 5 Report

Dear Author,

Present article can be accepted after major revision. You can find all comments in the attachment file.

Sincerely

Round 2

Reviewer 3 Report

Most suggestions were followed. Figures 12-17 are still marked with dots that in B&W are hard to distinguis.

Added graph for weight loss is appreciated, however regression from 0 is not linear for all materials.

Some references need check for example 42, “Arta”?

Reviewer 4 Report

I have read the answer of the authors and agree with publication of the article in this from.

Author Response

The next was revised as the english language editing is concerned 

Reviewer 5 Report

Dear Author,

Article in revised form can be accepted for publication.

Author Response

No further corrections were required